# Overlapping Machinery in Lysosome-Related Organelle Trafficking: A Lesson from Rare Multisystem Disorders

**DOI:** 10.3390/cells11223702

**Published:** 2022-11-21

**Authors:** Blerida Banushi, Fiona Simpson

**Affiliations:** Faculty of Medicine, University of Queensland, Diamantina Institute, Brisbane, QLD 4102, Australia

**Keywords:** lysosome-related organelles, endosomes, membrane trafficking, multisystem disorders, cargo-sorting, VPS33B, VIPAR, Arthrogryposis-Renal dysfunction-Cholestasis, Hermansky–Pudlak Syndrome, Griscelli syndrome

## Abstract

Lysosome-related organelles (LROs) are a group of functionally diverse, cell type-specific compartments. LROs include melanosomes, alpha and dense granules, lytic granules, lamellar bodies and other compartments with distinct morphologies and functions allowing specialised and unique functions of their host cells. The formation, maturation and secretion of specific LROs are compromised in a number of hereditary rare multisystem disorders, including Hermansky-Pudlak syndromes, Griscelli syndrome and the Arthrogryposis, Renal dysfunction and Cholestasis syndrome. Each of these disorders impacts the function of several LROs, resulting in a variety of clinical features affecting systems such as immunity, neurophysiology and pigmentation. This has demonstrated the close relationship between LROs and led to the identification of conserved components required for LRO biogenesis and function. Here, we discuss aspects of this conserved machinery among LROs in relation to the heritable multisystem disorders they associate with, and present our current understanding of how dysfunctions in the proteins affected in the disease impact the formation, motility and ultimate secretion of LROs. Moreover, we have analysed the expression of the members of the CHEVI complex affected in Arthrogryposis, Renal dysfunction and Cholestasis syndrome, in different cell types, by collecting single cell RNA expression data from the human protein atlas. We propose a hypothesis describing how transcriptional regulation could constitute a mechanism that regulates the pleiotropic functions of proteins and their interacting partners in different LROs.

## 1. Introduction

Endocytic pathways serve housekeeping functions in all cells allowing them to take up essential nutrients and enabling the breakdown and recycling of macromolecules as well as regulating temporo-spatial control of signal transduction cascades [1]. Specific adaptation of endosomes occurs in highly specialised cell types where membrane bound vesicles known as lysosome-related organelles (LROs) that share some features with lysosomes but are functionally and morphologically distinct, carry specific cargoes and confer unique properties to the harboring cell [1]. Immature LROs emerge from the Golgi Apparatus, early endosomes or Multivesicular bodies (MVBs) and mature by cargo delivery from tubulovesicular domains (Figure 1).

LROs include natural killer cell lytic granules, endothelial Weibel–Palade bodies, platelet-alpha and dense granules, notochord fluid-filled vacuoles, osteoclast granules, skin keratinocyte lamellar bodies and melanosomes, white blood cell basophil and azurophil granules and others [1] (Figure 1). Presynaptic vesicles in neurons could also be considered LROs but their biogenesis remains insufficiently understood [2]. Therefore, we did not cover this topic extensively and presynaptic vesicles are not included in Figure 1. These functionally diverse compartments share a variety of features with both lysosomes (lysosomal proteins, low pH) and secretory granules (sequestration, intracellular storage and regulated release of their luminal contents) [3,4]. LROs differ in their morphology, some closely resembling lysosomes with electron-dense material deposits whilst others having distinct features necessitated by their function [1,3,4,5,6,7] (Figure 1). In cells with specialised function, LROs carry and trigger the release of specific cargoes including inflammatory factors (mast cells), fibrinogen and von Willebrand factor (vWF) (platelets and megakaryocytes), haemostatic and pro-inflammatory factor (endothelial cells), surfactant for lung function (lung epithelial type II cells), lytic granules for the destruction of virally infected or cancerous cells (cytotoxic T lymphocytes (CTLs) and natural killer (NK) cells) [4] (Figure 1).

One of the of the most perplexing questions in the field of membrane trafficking is whether ubiquitous sorting pathways are used by different LROs in specialised cells or whether novel and cell-specific sorting pathways exist in different cells [3]. Many components of the molecular machinery required for the biogenesis of lysosome-related organelles are not conserved in yeast. In the last decade however, the study of multisystem disorders has played a fundamental role in the identification of the trafficking machinery required for LRO biogenesis and movement.

Here, we describe examples of trafficking machinery complexes repurposed among multiple LROs that are required for organelle motility or maturation (Figure 2). Not surprisingly, dysfunction in these proteins is associated with multisystem disorders affecting different organs. We outline their specific functions in each LRO and discuss how these functions might be linked.

We hypothesise that transcriptional regulation could constitute one of the several mechanisms that regulates the pleiotropic functions of proteins and their interacting partners.

## 2. Dysfunction of LROs Trafficking Proteins Is Associated with Rare Multisystem Disorders

The identification of genetic disorders that cause defects in multiple LROs clearly suggests that these organelles share common biogenic pathways. The study of these disorders has improved our understanding of the molecular machinery involved in the biogenesis of LROs [5,6,8]. They include several autosomal recessive disorders such as Hermansky–Pudlak syndrome (HPS) [9], Aarthrogryposis Renal dysfunction and Cholestasis syndrome (ARC) syndrome [10], Chediak–Higashi syndrome (CHS) [11], Griscelli syndrome (GS) [12] and other human primary immunodeficiency syndromes [13].

The clinical phenotype of patients suffering from these conditions can include at least one of the following features that can be related to defects in the formation or function of LROs: partial albinism may reflect defects in melanosomes, bleeding and inflammatory defects may reflect defects in endothelial WPBs, neurological defects may reflect malformation or dysfunctional secretion of synaptic vesicles, lung fibrosis may reflect defects in lamellar bodies, immunodeficiencies may reflect defects in the formation and release of cytolytic granules and/or the biogenesis of antigen presenting organelles [4]. The aforementioned LRO-related disorders share a clinical phenotype characterised by recurrent infections and immune system disorders that often cause the death of the affected patient. In Griscelli, Hermansky–Pudlak, and Chédiak–Higashi syndromes the clinical phenotype of ongoing infections is associated with albinism. This has provided valuable insight into the biogenesis of the secretory lysosomes from immune cells and melanocytes from endothelial cells and has suggested that common regulators are shared among LROs [14].

Mature LROs require specific components of the intracellular machinery (Rabs, SNAREs, effectors, motor proteins, Kinesin, Dynein) for their trafficking through microtubules and the actin cytoskeleton for delivery and ultimate secretion [15,16] (Figure 2). The trafficking of proteins to melanocytes and the mechanisms for sorting and secretion of the secretory lysosomes found in CTLs and NK cells has been reviewed elsewhere [5,14,17].

In mammalian skin, melanosomes are transported from the Golgi apparatus (and/or early endosomes) to the tips of melanocyte dendrites, and mature melanosomes are then transferred to neighboring keratinocytes [17,18] (Figure 1). Several Rabs and their effectors, such as Rab38, Rab32, VARP (effector of Rab38) [19,20], Rab 9a [21] and SNARE proteins (such as STX13) have been shown to regulate cargo transport to melanosomes [22].

The killing action of immune cells (e.g., NK- and T-cells) depends on their ability to interact with the target cell and to release perforin and granzymes that will destroy the target cell [23]. This ability is ‘centrally controlled’ by the immunological synapse [24], a physical ‘touchpoint’ that immune cells create by reorganising and protruding their membrane across the interaction site with the target cell. After binding, reorganization of both actin and microtubule cytoskeletons translocate the centrosome to the plasma membrane [25], while lytic granules move towards the microtubule organising center (MTOC). Recycling endosomes containing Rab11 and Munc13-4 merge with late endosomes (containing Rab27a and Rab7) which can bring exocytic traffic regulatory proteins needed for secretory granule release in immune cells [26]. Tethering of Munc13-4, Rab27, Rab11 structures with LAMP containing organelles in a process involving Qb SNARE vti1b, is required for release of secretory lysosome content in CTLs and macrophages [26,27].

## 3. Griscelli Syndrome Reveals Common Regulators Required for LRO Motility

Griscelli syndrome (GS, OMIM 214450) is a rare autosomal recessive disorder associated with hypopigmentation of the skin and hair and the presence of large clumps of pigment granules in hair shafts. Most patients also develop a severe immune disorder and/or severe neurological impairment without apparent immune abnormalities [12,28]. GS affects organelle motility and not maturation. Three forms of GS have been described.

GS1, 2 and 3 are caused by mutations in Myosin V [29,30], Rab27 [31,32,33], and Melanophilin [34] genes, respectively. The spectrum of clinical and cellular phenotypes in GS suggests that common molecular pathways are used by melanocytes, neurons, and immune cells in secretory granule exocytosis. In melanocytes Rab27a (GS2), Myosin Va (GS1) and Melanophilin (GS3) form a complex where the action of Rab27a and its effector Melanophilin is required for the motor Myosin Va to tether melanosomes to the actin-rich cell periphery for their transfer to keratinocytes [35,36] (Figure 3). The interaction between Rab27a, Myosin V and Melanophilin is therefore required for melanosome transport, and mutations in any of the genes coding these proteins leads to hypopigmentation of the skin and hair.

GS1 associates with neurological defects that lead to severe developmental delay and mental retardation in the early stage of development including ataxia and opisthotonos [29] (Table 1). GS2 is associated with immunodeficiency, and is characterised by absent delayed-type cutaneous hypersensitivity and an impaired immune cell function that resembles the accelerated phase or hemophagocytic syndrome (HS). Bone marrow transplant represents the only cure for these patients [30,37]. GS3 is associated with typical hypopigmentation of GS with no immunological or neurological manifestations [38]. Mouse models for GS1, 2 and 3, (dilute [39], ashen [40], and leaden [41] respectively), resemble the phenotype of their human counterparts.

At a cellular level, an accumulation of melanosomes in melanocytes is observed with abnormal transfer of melanin granules to keratinocytes [12,28,32,33,40,41], while impaired natural-killer cell function, uncontrolled lymphocyte and macrophage activation is observed in GS2 [40] (Table 1). Studies suggest that neurological defects in GS1 can be potentially caused by the inability of the endoplasmic reticulum to move into the dendritic spines [29] (Table 1).

## 4. GS1 and MYH9RD Are Caused by Defective Myosin-Mediated LRO Trafficking

Mutations in MyosinVa in GS1 are associated with both neurological and pigmentation defects suggesting that Myosin-Va is required for melanosome and neuronal vesicle transport [57].

MyosinVa is a member of the Myosin V family of actin-dependent motors that binds to cytoskeleton filaments through an N-terminal head domain, and to its cargo through the C-terminal globular tail [58]. MyosinVa is expressed at high levels, but not exclusively, in the brain [59] and this is related to its role in small synaptic vesicle trafficking. Other studies have shown that MyosinVa forms a complex with Rab27a and the effector MyRIP (myosin- and Rab27a-interacting protein) in endothelial cells where it regulates the acute release of von-Willebrand factor [60] (Figure 3). This explains the platelet defects in the Ashen mice (*Rab27a*^–^), the mouse model for GS2, that exhibits increased bleeding and a reduction in the number of platelet dense granules (Table 1) [40,61].

Myosin Va also regulates trafficking of dense core vesicles (packed with hormones) in hormone-producing cells such as the adrenal gland or the pancreas [62].

Myosin Va and Melanophilin are not expressed in cytotoxic cells, therefore dysfunction in these proteins in GS1 and GS3 do not affect the secretion of lytic granules, suggesting that different classes of effector proteins are involved in trafficking of lytic granules in immune cells [41]. Myosin IIa has been shown to regulate trafficking of lytic granules by functioning at the level of the immunological synapse of NK cells [63] and to regulate the secretion of vWF from the endothelial Weibel-Palade bodies (WPBs) [64] (Figure 3). Mutations in *MYH9* gene (Myosin IIa) are associated with a number of autosomal dominant disorders, collectively called MYH-9–related disease (MYH9RD) where patients are characterised by congenital macrothrombocytopenia with mild bleeding tendency and they may develop kidney dysfunction, deafness and cataracts later in life [65]. At a microscopic level MYH9RD patients present with thrombocytopenia, enlarged platelets and Döhle-like bodies (basophilic inclusion bodies that may be found in individuals with infection) in the cytoplasm of neutrophils (Table 1).

## 5. Rab27a and Its Effectors in LRO Trafficking

GS2 mutations in Rab27a affect both the secretion of cytotoxic granules and melanosomes [31,32,33,40] (Table 1). In CTL and NK cells, Rab27a regulates the release of lytic granules through its interaction with Munc13-4 mediating vesicle docking and priming (Figure 3) [66]. Mutations in Munc13-4 *(UNC13D* gene) cause FHL type 3, a form of familial hemophagocytic lymphohistiocytosis characterized by lytic granules that dock but fail to fuse with the plasma membrane at the immune synapse, resulting in patients that present typical features of FHL such as early onset of overwhelming activation of T lymphocytes and macrophages [67] (Table 1). Munc13-4 is also involved in dense core granule secretion in platelets [60] leading to severe defects in the release from dense and α-granules of platelets of Unc13d(Jinx), the mouse model for FHL3 with prolonged tail-bleeding times [43]. Although bleeding has generally not been associated with patients affected by FHL3, abnormalities in platelet secretion could also be indicative of FHL3 [43,68].

The Rab27a subfamily regulates various exocytotic pathways using multiple organelle-specific effector proteins [69,70]. It is a key player in the release of secretory vesicles from melanocytes [32,33], CTLs [71] and neutrophils (through its effector Munc 13-4) [72], it regulates the secretion of insulin granules (that are not LROs) in pancreatic beta cells through its interaction with Syntaxin 1a, Munc18-1 and its effector granuphilin [73], it regulates the release of vWF in endothelial cells [60] and the dense core granule secretion in platelets through its interaction with Munc13-4 [74] or Syntaxin 1a/Munc18-1/Slp4 [75].

Some of the Rab27a effectors and their cell specificities are summarised in Figure 3. The molecular and clinical phenotype of GS1 and 2 patients and mouse models (Table 1), suggest a close relationship between platelet dense granules, melanosomes of melanocytes and secretory lysosomes of CTLs, all mediated by Rab27a. Other putative Rab27 effector proteins, named exophilins or Slp/Slac2, share the conserved N-terminal Rab27-binding domain and show Rab27-binding activity in vitro or when overexpressed in cell lines [69]. A Rab27a/Slp3/Kinesin-1 complex was shown to regulate trafficking of cytotoxic granules in CTLs [76]. Additionally, Rab27a regulates secretion of azurophilic granules [77] and eosinophil degranulation [78] but is has been shown to play an inhibitory role in regulating degranulation from mast cells [79]. A Rab27b/Slp3/Kinesin-1 is responsible for mast cell degranulation [80].

Rab27a is not expressed in osteoclasts but only in precursor cells [81]. During osteoclast differentiation Rab27a regulates the transport of cell surface receptors modulating multinucleation and LROs [82]

Rab27 has since been shown to regulate androgen-dependent secretion in prostate epithelium secretory cells via synaptotagmin-like protein (SLP1) effector [83] and the SLP1/Rab27 interaction is also required for the anterograde transport of tropomyosin receptor kinase (TrkB) in axons [84]. Rab27 is also involved in the secretion of granules that are not LROs such as zymogen granules in pancreatic acinar cells [85], mucus granules in gastric surface mucous cells [86] and glucagon granules in pancreatic alpha cells [87].

Other Rab27 effectors and their role in the secretory pathways in different LROs is reviewed in [88].

By binding to different effector proteins, Rab27a regulates trafficking and secretion of different LROs in different cell types. Common effector proteins are however shared among different LROs suggesting that the protein complexes they are part of, rather than the effectors themselves is what gives specificity to LRO trafficking (Figure 3B).

## 6. CHS Is Caused by Impaired LYST-Mediated LRO Fission

Chediak–Higashi Syndrome (CHS) is a rare autosomal recessive disorder characterized by severe immunodeficiency, oculocutaneous albinism, bleeding tendencies, recurrent life-threatening infections, varying neurologic problems and lymphohistiocytosis [44,89].

The hallmark of this lethal multisystem disorder is the presence of giant intracellular granules that cannot be secreted in different cell types including lysosomes, melanosomes, cytolytic granules and platelet dense bodies, as a result of decreased fission of lysosomes and LROs [89,90].

The defective gene, *LYST* or *CHS1*, encodes a 3801-residue protein that is highly conserved throughout evolution [11]. In yeast the LYST homolog Bph1 is a Rab5 effector and prevents Atg8 lipidation at endosomes, suggesting that LYST contributes to the maintenance of endosomal functions [91]. In a Dictyostelium model, the orthologous of LYST, lvsB functions by antagonizing Rab14-dependent homotypic lysosomal fusion between lysosomes and LROs [92]. LYST also plays a role in phagolysosomal maturation by controlling toll-like receptors (TLR)3- and TLR4-induced endosomal TRIF proinflammatory signaling pathways [93]. These data demonstrate how regulatory mechanisms in membrane trafficking can directly affect specific immune receptor signaling pathways, that may ultimately explain the immunodeficiency associated with CHS [93].

## 7. Defects in LRO Biogenesis: The Hermansky–Pudlak Syndrome

The study of the molecular basis of Hermansky–Pudlak Syndrome (HPS) has greatly contributed to our understanding of the mechanisms by which lysosomes and related organelles are formed. HPS is a group of autosomal recessive diseases (OMIM 2033000) all of which exhibit oculocutaneous albinism reflecting defects in the biogenesis of melanosomes, and excessive bleeding and bruising as a result of defects in the biogenesis of platelet-dense granules [9,94,95]. These rare genetically heterogeneous set of related autosomal recessive conditions affect the ubiquitous molecular machinery required for the biogenesis of different LROs in different cell types. These genes are ubiquitously expressed in all cells suggesting additional cell-type specific functions in cells that lack LROs.

Mutations in 10 different genes leading to abnormal protein products cause 10 different types of HPS in humans (HPS1-10) (Table 1).

In addition to the common HPS phenotypical features, patients affected from different HSP subtypes suffer from progressive lung fibrosis as a result of defects in the biogenesis of lamellar bodies in type II lung epithelial cells (HPS2 or HPS4) [96,97,98], the accumulation of a lipid-protein complex called ceroid lipofuscin [99,100], or recurrent bacterial and viral infections due to impaired cytotoxic T and NK cell activity (HPS2) [101,102]. A summary of the clinical and cellular phenotype for each HPS subtype is reported in Table 1. HPS is caused by mutations in any of the genes coding subunits of different multi-protein complexes that include the adaptor protein3 (AP3) [103] and the biogenesis of lysosome-related organelles complex (BLOC)-1, -2 and -3 (Table 1). The disease spectrum is very similar in patients with mutations in different subunits of the same protein complex (Table 1). Mutations in additional genes cause HPS-like phenotypes in mice and these include the gene encoding Rab32 and Rab38 or the vacuolar protein sorting (VPS)33a, one of the two mammalian orthologs of yeast Vps33 and subunits of the homotypic fusion and vacuole protein sorting (HOPS) and the class c core vacuole/endosome tethering (CORVET) complexes [104,105]

## 8. AP3 Sorts Different Cargoes in Different LROs and Its Deficient Function Associates with HPS2 and HPS10

AP3 is a stable heterotetramer consisting of four adaptin subunits δ, β3, μ3 and σ3 [103,106] that are highly conserved from yeast to humans [107,108].

In yeast, the AP3 complex functions in cargo-selective protein transport from the Golgi to the vacuole [109] suggesting that AP3 is a pleiotropic protein with a high number of interacting partners that act at different intracellular compartment levels.

In humans, AP3 plays a unique role in vesicle cargo sorting from early endosomes to lysosomes or LROs and similarly to the yeast homologue, it plays a versatile role in binding different cargoes through the μ3A subunit for specific deliveries to specific LROs (Figure 4) [110].

Mutations in genes encoding two different ubiquitous AP3 subunits, β3A and δ, in humans cause, respectively HPS2 and HPS10 that are a reflection of cargo missorting in different LROs [111,112] (Table 1). The phenotypes of HPS2 and HPS10 are highly overlapping with the exception of neurological defects specific to HPS10 [113]

Two forms of the AP3 heterotetramer complex exists, one ubiquitous one neuronal-specific. The AP3δ subunit (affected in HPS10 patients) is essential for both forms and cannot be compensated, but the AP3β3A subunit, (affected in HPS-2 patients) is substituted by AP3β3B in the neuron-specific AP3 complex and can therefore be compensated in neurons but not in other tissues [113]. This explains the difference in the neurological phenotype (including neurodevelopmental delay, epilepsy and a hearing disorder) between HPS2 and HPS10.

In neurons the AP3 pathway is required for sorting of specific cargoes from the early endosomes to synaptic vesicles [114,115] Examples are the zinc transporter ZnT-3, or the gamma-aminobutyric acid (GABA) transporter [115,116] (Figure 4). AP3 deficient mice showed a reduction of vesicles carrying GABA that led to deficiency in synaptic transmission with the mice suffering from spontaneous recurrent epileptic seizures [116].

CD63, LAMP-1, and LAMP-2 have been shown to accumulate in the cell surface of fibroblasts from HPS2 patients suggesting that AP3 is required for targeting these proteins to the lysosome [117] (Figure 4).

In melanocytes, AP3 sorts tyrosinase (TYR), a membrane protein responsible for the first step in melanin production, from the early endosomes to maturing melanosomes [17,118] (Figure 4) and both HPS2 and HSP10 patients or the mice models (Table 1) are characterised by hypopigmentation and oculocutaneous due to lack of TYR in melanosomes, and its accumulation in the early endosomes [113,118,119].

Among other LROs affected in HPS, the lamellar bodies (LB) in alveolar type 2 cells are enlarged and accumulate phospholipids. Kook et al. have shown that AP3 is required for the delivery of the soluble enzyme peroxiredoxin 6 (PRDX6) to LB through binding to the transmembrane protein LIMP-2/SCARB (Figure 4) [120]. The LBs from the *pearl* mouse (HSP2 model) lack PRDX6 and accumulate phospholipids [120].

The immune system of HPS2 and HPS10 patients and their corresponding mouse models *pearl* and *mocha,* is compromised leading to recurrent bacterial and viral infections (Table 1) [47,53]. HPS2 patients suffer from cyclic neutropenia, characterized by recurrent episodes of low levels of neutrophils, and this has been linked to missorting of neutrophil elastase to granules [121,122]. AP3 is therefore required for sorting neutrophil elastase but the mechanism of this interaction is still not clear, whether this is direct (with elastase being transmembrane) or indirect, through a transmembrane linker protein similar to that described for PRDX6 [120,123]. Additional factors that contribute to the immunodeficiency in HPS2 and HPS10 are defects in the recruitment of TLRs from the endosomes to maturing phagosomes where TLRs are required for proinflammatory TLR signalling and antigen presentation of phagocytosed antigens to CD4+ T cells, following a bacterial or viral infection [124,125]. In addition to TLRs, AP3 is required for the trafficking of CD1b that presents glycolipids to natural killer T (NKT) cells [126,127]. The mouse homologue (CD1d) has been shown to interact with AP3 and mediate its trafficking to late endosomes/lysosomes of NKT cells (Figure 4). Levels of CD1d in cells from AP3 deficient mice are decreased in the late endosomes but increased on the cell surface. This results in a decrease of NKT cells in the thymus, spleen and liver that correlates with the HPS2 and HPS10 patient phenotypes that also have reduced numbers of NKT cells leading to current infections [126].

AP3 plays an essential role also in the biogenesis and release of secretory granule in mast cells [128]. Disruption of cargo proteins delivery to dense granules could explain platelet deficiency and excessive bleeding in HPS2 and HPS10, but these proteins have yet to be identified.

## 9. BLOC1, -2 and -3 Sort Different Cargoes in Different LROs and Its Deficient Function Associates with HPS1 and HPS3-9

AP3 and BLOC1-3 and are part of the machinery that mediates vesicular transport of integral membrane proteins to lysosomes and LROs. AP3 and BLOC1 interact in neurons and melanosomes to sort specific cargoes into synaptic vesicles or melanosomes, respectively [129,130]. However, depending on the cargo, the transport of BLOC1 to LROs can be AP3 dependent (transporter OCA2 [131]) or independent (ATP7A [132] and the melanogenic enzyme TYRP1 [118]) (Figure 4 and Figure 5). Patients with mutations in BLOC1 (HPS7, HPS8 and HPS9) have milder symptoms of the disease without immune or lung dysfunctions (Table 1).

BLOC1 (in addition to AP3) is involved in cargo sorting from the early endosomes to melanosomes that correlates with the oculocutaneous albinism phenotype in HPS7, HPS8 and HPS9 patients (Table 1). These cargoes include OCA2 [130,131], TYRP1 [129,133], ATP7A [132] and a small amount of TYR (that is not trafficked through the AP3 dependent route) [134], which are all involved in different steps of melanin synthesis (Figure 5). BLOC1 interaction with ATP7A is also required in neuronal cells for correct copper metabolism and BLOC1 loss of function is associated with risk of schizophrenia [135].

BLOC1 and AP3 are involved in sorting different cargoes (such as VAMP7-TI and PI4KIIα, Figure 5), to synaptic vesicles in neuronal cells [114] and this process can be specific to the brain region [136]. In agreement with these findings, BLOC1-deficient mice have abnormal kinetics of neurotransmitter release [137]

BLOC1 deficient mice demonstrated altered targeting of LAMP1 similarly to AP3-deficient mice [138] (Figure 5). Indeed BLOC1 regulates the association of the phosphatidylinositol-4-kinase type II alpha (PI4KIIα) with AP3 [139]. BLOC1 regulates endosomal sorting of PI4KIIɑ in several cell types (including neurons, Figure 5) through its interaction with the WASH (the WASP and SCAR homologue) complex [140]. BLOC1 dependent transport from the early endosomes to melanosomes or LROs occurs via distinct tubular transport carriers that have similar features to recycling endosomes [141,142]. BLOC1 is involved in transferrin recycling in HeLa cells [142] and is specifically required for the formation of these tubular structures [142]. In addition to the WASH complex, BLOC1 interacts with the actin and microtubule cytoskeleton [142,143,144], and SNARE complexes (such as Syntaxin13 and SNAP-25) to facilitate membrane fusion and effect cargo transport [141,142,145] (Figure 5B). For instance, BLOC1 interaction with Syntaxin2 and TSG101 (an endosomal sorting complex required for transport subunit-I) is required for EGFR trafficking to the lysosomes for its degradation [146] (Figure 5). Bowman et al. have shown that binding of either AP3 to VAMP7 or BLOC1 to STX13 is both necessary and sufficient to support VAMP7 sorting and melanogenesis [147]. This mechanism of redundancy in the SNARE sorting step provides insight into how SNAREs in different locations within the endolysosomal system can maintain specificity in membrane fusion reactions [147].

BLOC2 is mutated in HPS3,5 and 6 (Table 1) and similarly to patients with mutations in BLOC1, HPS patients with BLOC2 mutations have milder symptoms of the disease that do not include immune or lung dysfunctions but the patients have a severe tendency to bleed [119,148,149] and this has been associated with defects in the maturation and secretion of Weibel-Palade bodies in endothelial cells [150]. Although the function of BLOC2 in LRO biogenesis is not completely understood, BLOC1 is known to interact with BLOC2 and participate in the same pathway for cargo sorting from early endosomes to melanosomes [129,130,131,132,133,134] and targeting the formation of recycling endosomal tubules for their contact with melanosomes [151] (Figure 5).

BLOC3 has GEF activity for Rab32 and Rab38 GTPases [152] (that as mentioned cause HPS-like phenotypes in mice) that regulate cargo delivery to nascent melanosomes including TYRP1 and TYR cargo delivery (Figure 5) [105]. These data likely explain the oculocutaneous albinism observed in patients with mutations in BLOC3 (HPS1 and HPS4) (Table 1). BLOC3 is also a Rab9 effector, although this activity is not required for melanogenesis [153]. In addition to melanosomes Rab32 and Rab38 regulate cargo delivery to platelet dense granules [154,155], lamellar bodies [156,157] and notochord vacuoles [158] but these mechanisms are not yet fully known.

## 10. The CHEVI Complex in Arthrogryposis, Renal Dysfunction and Cholestasis Syndrome

Arthrogryposis, Renal dysfunction and Cholestasis (ARC) syndrome (OMIM #208085 and #613404) is a rare autosomal recessive multisystem disorder with clinical presentation that includes arthrogryposis, renal tubule dysfunction and neonatal cholestasis [159]. Additional features include ichthyosis, sensorineural deafness, central nervous system malformation, abnormal platelet α-granule biogenesis, recurrent infections and severe failure to thrive [159] (Table 1). Patients affected by ARC syndrome do not survive past their first year due to recurrent infections, dehydration, acidosis or uncontrolled bleeding, although a few patients with an attenuated phenotype have been reported to survive until childhood [160,161]

ARC syndrome is caused by germline mutations in the genes *VPS33B* encoding for VPS33B (75% of all ARC syndrome cases) and *VIPAS39* encoding for VIPAR (25% of all ARC syndrome cases) [10,159,160,161,162,163]

VPS33B and VIPAR are ubiquitously expressed in all cells. Like VPS33A, VPS33B is one of the two Sec1/Munc18 (SM) protein homologues of the yeast protein Vps33p, that interact with SNAREs to regulate vesicular fusion events [159,164]. However, VPS33A and VPS33B have separate functions and cannot compensate for each other [163,165,166]. The yeast Vps33p and the mammalian homologue VPS33A are part of the class C core vacuole/endosome tethering (CORVET) and homotypic fusion and vacuole protein sorting (HOPS) complexes that regulate cargo delivery to the vacuole and lysosome, respectively [167,168,169,170]. Similarly, the yeast Vps16p, and its mammalian homologue VPS16 are another component of the yeast and mammalian HOPS and CORVET complexes, which directly interact with Vps33p and VPS33A, respectively [10]. However, neither VPS33B nor VIPAR are part of the mammalian CORVET or HOPS complexes but instead form part of a separate complex with different functions, the class c homologs in endosome-vesicle interaction (CHEVI) complex [166,170,171,172].

Whether the CHEVI complex is part of bigger multiprotein complex similarly to HOPS and CORVET remains unknown, but studies have suggested that CHEVI is a small complex including just VPS33B and VIPAR [166]. Multiple studies have implicated the CHEVI complex in specific and context-dependent trafficking of a diverse range of cargo to varied organelle compartments that differ among specialised cell types [10,163,166,171,173] (Figure 6).

The CHEVI complex has been shown to act at the recycling endosomes in association with Rab11a in polarized epithelial cells [10]. Analysis of liver biopsies from ARC patients showed mislocalisation of specific apical membrane proteins that undergo recycling (e.g., bile salt export pump and CEA), from the hepatocyte apical membrane [10,163], whereas other apical membrane proteins (such as multi-drug resistance protein 2 (MRP2)) that do not undergo recycling are correctly localized [54]. These findings correlate with the severe cholestasis in ARC patients and murine models and are a result of increased levels of alkaline phosphatase and bile acids in the blood due to failure to deliver these molecules to the bile [54,159]

VPS33B has been shown to interact with Rab11a [10] but it is still unknown whether the CHEVI complex directly binds to its specific cargoes for delivery in hepatocytes (Figure 6). Interestingly Vps33b liver specific knockout (with a liver-specific *Vps33b* gene mutation) demonstrated focal steatosis in their liver suggesting defects of lipid metabolism [54]. Although the role of the CHEVI complex in lipid metabolism is yet to be established, lipid droplets have been observed in hepatocytes from HPS mutant mice and HPS proteins are likely to be involved in lipid metabolism in hepatocytes, thereby affect lipid content in plasma [174].

Bleeding defects associated with the ARC syndrome and mouse models result from decreased numbers of α-granules and defective platelet aggregation [55,175]. Megakaryocytes from Vps33b deficient mice have smaller α-granule-like vesicles as well as abnormal vacuolar structures, possibly due to defective MVB maturation into α-granules [55]. The α-granule proteins such as vWF were also reduced in the MVBs, suggesting defects in trafficking of proteins to MVBs [55,176]. VPS33B has been shown to directly interact with α-tubulin, SEC22B and the integrin β-subunit for the transport of proteins to MVBs and MVB maturation into α-granules [176,177] (Figure 6). Binding of VPS33B with α-tubulin reflects the importance of the association of tethering complexes with the cytoskeleton for vesicle transport, similarly to the previously described BLOC1 (Figure 5b) [142]

In addition to α-granules, different LROs such as the lamellar bodies are also abnormally formed in ARC patients and in *Vps33b* and *Vipar* mouse models explaining the ichthyosis with dry, scaly and irritated skin [159,178,179]. In murine models the LB cargo, Kallikrein-related peptidase 5 (KLK5) was abnormally expressed in the epidermis and additionally the lipid deposition in the stratum corneum was also affected suggesting a role for the CHEVI complex in cargo trafficking to LB and lipid metabolism (as mentioned for the liver) [178]. It has been suggested that CHEVI-mediated trafficking to the lamellar bodies occurs via Rab11a recycling endosomes [10,171,180] (Figure 6).

The CHEVI complex is ubiquitously expressed in all cells, therefore additional roles for this complex have been postulated for cells that do not have LROs. In mouse kidney cells the CHEVI complex has been shown to be required for trafficking of the collagen-modifying enzyme, lysyl hydroxylase 3 (LH3) in a Rab10/Rab25 dependent pathway, from the TGN to newly identified compartments collagen IV carriers (CIVCs) where collagen IV is post-translationally modified [56]. Banushi et al. propose that LH3 trafficking mediated by CHEVI occurs through a yet unidentified transmembrane protein similarly to the aforementioned AP3 mediated trafficking of the neutrophil elastase or PRDX6.

Whether CIVCs can be considered a LRO it is still unknown and further studies are required to characterise this novel compartment [56]. This study however explains several collagen-related defects associated with phenotypes of ARC patients, murine models and cell lines [56,178,181]. Additionally, the authors demonstrate an epithelial to mesenchymal transition (EMT) phenotype with abnormal collagen formation and secretion in Vps33b and Vipar knockdown mouse kidney cell lines [56] that explains data attributing a tumour suppressor role of VPS33B in several human malignancies [182,183,184]

## 11. Transcriptional Regulation of Proteins within the Same Complex Could Constitute a Cell-Specific Mechanism for Protein Function Versatility

Proteins within the same complex are often encoded by genes that are regulated by the same transcription factors, they often exhibit expression coherency and include synergistic transcription factors pairs [185,186,187]. This synergic expression at mRNA and protein level has been showed for the members of the CHEVI complex VPS33B and VIPAR [10,56,162,188]. Firstly, reduced expression of VPS33B were detected in approximately 25% of patients without known *VPS33B* mutations [162] and this was later shown to be caused by mutations in *VIPAS39* that accounted for all cases of ARC where *VPS33B* mutations were excluded (~25%) [10]. At a protein level, Banushi et al. observed a largely improved recombinant expression yields for VPS33B and VIPAR when the two proteins were coexpressed in HEK293 cells compared with production of single proteins [56]. A further example demonstrating a functional relevance of the synergic co-regulation of VPS33B and VIPAR, was shown by Western blot and immunofluorescent analysis of conditionally immortalized human glomerular cells that demonstrated a greater expression of both VPS33B and VIPAR in podocytes and glomerular endothelial cells, that constitute the filtration barrier of the kidney, but not in the mesangium [188].

Protein complexes involved in overlapping machineries in LRO trafficking could exhibit cell-specific transcriptional regulation, in relation to their specific function and interacting partners in the LRO of that cell type.

We analysed the expression of *VPS33B* and *VIPAS39* in multiple cell types by collecting single cell RNA expression from the human protein atlas (https://www.proteinatlas.org/ accessed on 18 July 2022).

*VPS33B* and *VIPAS39* exhibit similar expression in (1 < fold change < 2) in the majority (~65%) of analysed cell types (Figure 7A). Given the aforementioned correlation between *VPS33B* and *VIPAS39* expression at a protein and RNA level [10,56,162,188], we hypothesise that the similar stoichiometric expression could be related to their function as a separate binary complex, which has already been proposed from literature [166]. Although analyses of correlation within single cells are required to test this hypothesis, the high number of cell types analysed could suggest that this outcome is very unlikely due to chance. In ~23% (Figure 7B) of the cell types a 2–4-fold-change between *VPS33B* and *VIPAS39* expression is observed. In ~12% of the cell types *VPS33B* and *VIPAS39* show 5-fold or higher difference in protein expression (Figure 7C).

VPS33B and VIPAR deliver different cargoes in collecting duct cells and hepatocytes (Figure 6). Their relative expression also differs among these cell types (Figure 7A,B, respectively) and could constitute a mechanism of functional regulation and interaction with different binding partners. However, post- transcriptional processes are not taken into consideration in this analysis, and should be analysed in future work aimed at testing our hypothesis. A fold change > 5 between *VPS33B* and *VIPAS39* is observed in melanocytes, and *VIPAS39*′s RNA expression is null (Figure 7C) which suggest that different interacting partners could be associated with VPS33B in these cells and mediate a different function in organelle trafficking. As mentioned previously, melanosome biogenesis is mediated by VPS33A and not VPS33B, and *VPS33A* mutations impair melanosome biogenesis in the Hermansky-Pudlak syndrome. Therefore, the VPS33B-VIPAR complex is not involved the biogenesis of melanosomes in melanocytes, and this could be due to its transcriptional regulation, further supporting our hypothesis. Further analysis of the *VPS33A* and *VPS16* expression (that associate as part of the HOPS complex) could provide insights into the transcriptional regulation of these proteins in relation to their cell-specific function.

## 12. Conclusions

The study of multi-system disorders has led to the identification of common mechanisms of LRO biogenesis and disease phenotypic outcomes. A spectrum of clinical phenotypes including albinism, lung fibrosis, immunodeficiency, ataxia, opisthotonos and congenital macrothrombocytopenia are linked with mutations in proteins involved in LRO biogenesis. The common proteins involved in LRO pathways have been identified and include Rab27a and its effectors, Myosins, LYST, AP3, BLOC1-3, and the CHEVI complex. Analysis suggests that protein complexes involved in overlapping machineries in LRO trafficking could exhibit cell-specific transcriptional regulation, in relation to their specific function and interacting partners in the LRO of that cell type. An understanding of the relative balance of protein components leading to differential functions in the biogenesis of different LRO types may allow for differential pharmacology to be identified in drugging targets for each complex LRO disorder.

## Figures and Tables

**Figure 1 cells-11-03702-f001:**
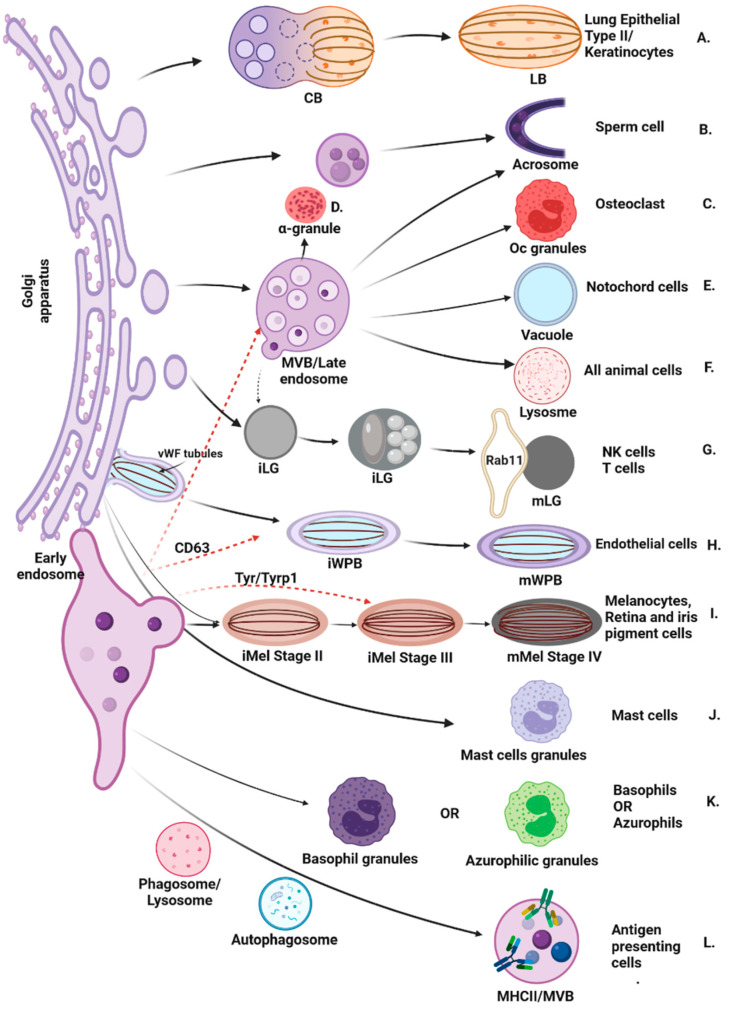
Model for biogenesis of mammalian LROs in different cell types. Shown are models for the biogenesis of LROs in specific cell types. A. Composite body (CB) mature into lamellar body (LB) in lung epithelial type II. B. The acrosome sac in sperm cells is formed and enlarged by the continuous fusion of Golgi-derived vesicles. C. Osteoclast (Oc) granules are generated in osteoclast from MVBs. D. Alpha granules are generated in megakaryocytes from MVBs. E. Notochord vacuoles are generated in notochord cells from megakaryocytes. F. Lysosome generate from MBVs. G. Immature LGs (iLGs) in NK or T-cells derive by fusion of MVBs with dense core structures and upon stimulation fuse with Rab11-positive recycling endosomes to form mature LGs (mLGs). H. Immature Weibel–Palade bodies (iWPB) derive from tubules in the TGN of endothelial cells to form mature WPBs (mWPB). I. Melanosomes (Mel) generate from the Golgi and/or early endosomes through different stages (II-IV) where specific enzymes (eg. Tyr(tyrosinase)/Tryp(tyrosinase-related protein-1) are delivered from the early endosomes. J. In mast cells, granules containing different contents are generated from the Golgi. K. Basophil and azurophilic granules generate from the early endosomes in basophils and azurophils, respectively. L. MHC II-related proteins are synthesized in the endoplasmic reticulum and directed to MVBs via the TGN.

**Figure 2 cells-11-03702-f002:**
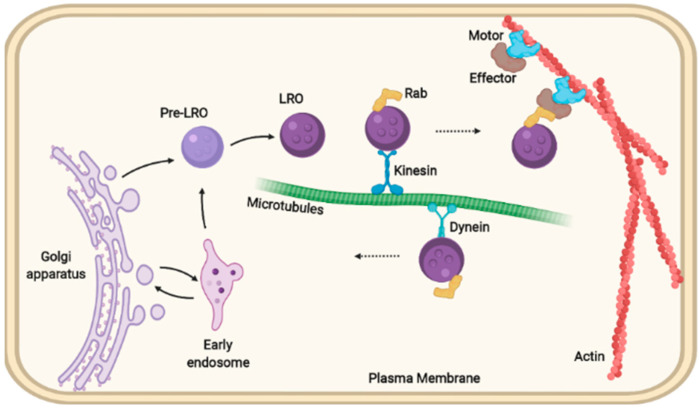
Intracellular delivery of lysosome-related organelles. Mature lysosome-related organelles (LROs) require specific components of the intracellular machinery (Rabs, SNAREs, effectors, motor proteins, Kinesin, Dynein) to assist their trafficking through microtubules and the actin cytoskeleton for delivery and secretion.

**Figure 3 cells-11-03702-f003:**
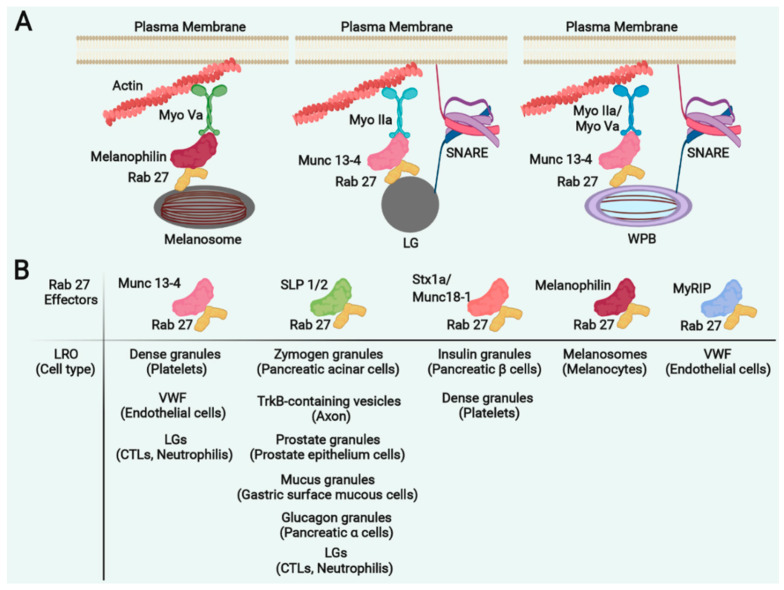
Common regulators of LRO trafficking in different cell types. (**A**). In melanocytes, Myosin Va(Myo Va) forms a complex with Rab27a and its effector Melanophilin, required to tether melanosomes to the actin cytoskeleton. In CTL and NK cells, Myosin IIa forms a complex with Rab27a and Munc13-4, which regulates the release of lytic granules. In endothelial cells, Mysin IIa (or MyosinVa) forms a complex with Rab27a and Munc13-4, which regulates the acute release of von-Willebrand factor from Weibel-Palade bodies (WPB). (**B**). Multiple effectors of Rab27a in different LROs of different cell types. Munc 13-4, mammalian uncoordinated 13-4; Slp, synaptotagmin-like protein; Stx1a, Syntaxin 1a; MyRIP, myosin VIIA and Rab interacting protein; TrkB, tropomyosin receptor kinase.

**Figure 4 cells-11-03702-f004:**
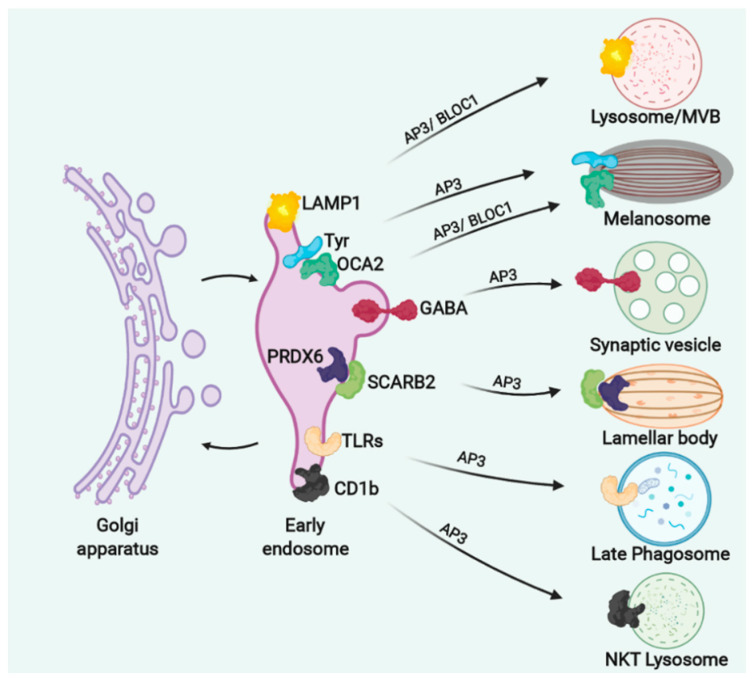
AP3 sorts different cargoes to lysosomes and LROs. AP3 is required for sorting of different cargoes from the early endosomes: LAMP1 to lysosomes, tyrosinase (TYR) and transporter oculocutaneous albinism 2 (OCA2) to maturing melanosomes, gamma-aminobutyric acid (GABA) to synaptic vesicles, peroxiredoxin 6 (PRDX6) to lamellar bodies through binding of the transmembrane protein LIMP-2/SCARB, toll-like receptors (TLRs) to maturing phagosomes and CD1b to natural killer T (NKT) cells. BLOC1 is involved together with AP3 in the delivery of LAMP1 to lysosomes and OCA2 to maturing melanosomes.

**Figure 5 cells-11-03702-f005:**
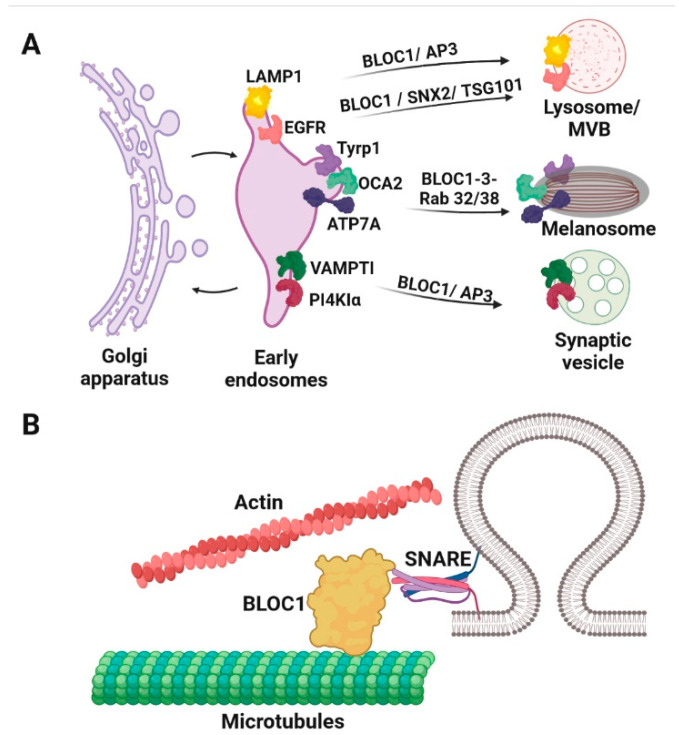
BLOC 1–3 sorts different cargoes to lysosomes and LROs. (**A**). BLOC1 and AP3 are involved in sorting different cargoes from early endosomes, including LAMP1 to the lysosomes as well as VAMP7-TI and PI4KIIα to synaptic vesicles in neuronal cells. Guanine-nucleotide exchange factor (GEF) activity of BLOC3 for Rab32 and Rab38 GTPases is required for TYRP1, OCA2 and ATP7A cargo delivery from the early endosomes to mature melanosomes. (**B**). BLOC1 is associated with SNARE complexes, microtubules and the actin cytoskeleton.

**Figure 6 cells-11-03702-f006:**
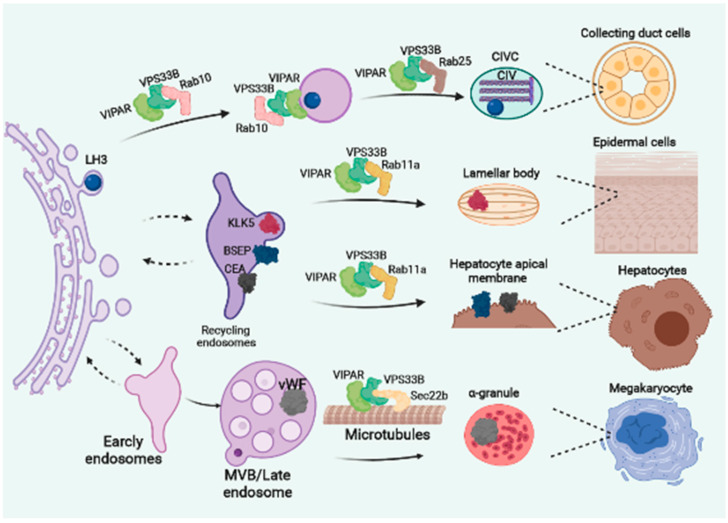
CHEVI complex regulates delivery of different cargoes in different intracellular compartments. The VPS33B/VIPAR protein complex delivers lysyl hydroxylase 3 (LH3) to collagen IV carriers (CIVC) in Collecting duct cells, Kallikrein-related peptidase 5 (KLK5) from the recycling endosomes to lamellar bodies in epidermal cells, bile salt export pump (BSEP) and the carcinoembryonic antigen (CEA) from the recycling endosomes to the hepatocyte apical membrane in hepatocytes and von Willebrand factor (vWF) from multivesicular bodies to α-granules through the interaction with α-tubulin and SEC22B.

**Figure 7 cells-11-03702-f007:**
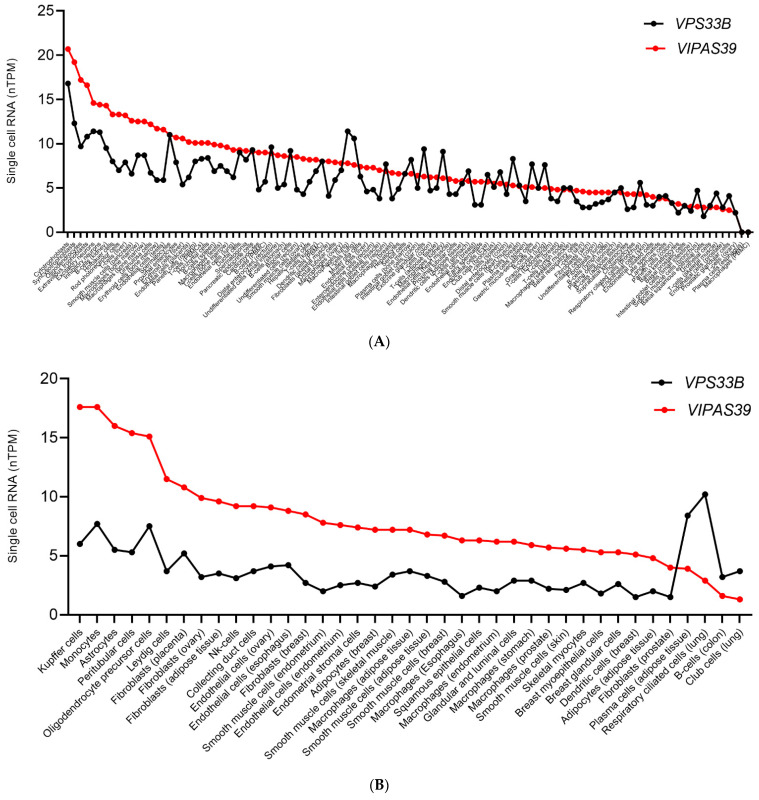
Single cell RNA expression of *VPS33B* and *VIPAS39* in different cell types. RNA levels for each gene are measured in normalised transcripts per million (nTPM) based on RNA-seq data from the Human Protein Atlas (https://www.proteinatlas.org/ accessed on 18 July 2022). (**A**). 1–2-nTPM fold-change between *VPS33B* and *VIPAS39* (1 < fold change < 2). (**B**). 2–4-nTPM fold-change between *VPS33B* and *VIPAS39* (2 ≤ fold change ≤ 4). (**C**). Greater than 5- nTPM fold-change between *VPS33B* and *VIPAS39* (fold change ≥ 5).

**Table 1 cells-11-03702-t001:** Multisystem disorders associated with mutations in genes involved in vesicular trafficking to LROs. Mutated gene, clinical and cellular phenotype as well as the correspondent mouse model are reported. * Abnormalities observed in the mouse model. Abbreviations: *MYH9*, Myosin-9; MYH9RD, MYH9-related thrombocytopenia; FHL3, Four and a half LIM domains protein 3; HPS, Hermansky–Pudlak Syndrome; ARC, Arthrogryposis, Renal dysfunction and Cholestasis.

Syndrome	Mutations	Clinical Phenotype	Cellular Phenotype	Mouse Model
Griscelli Type 1	*MYO5A*	Hypopigmentation of skin and hair, Neurological disorder	Abnormal accumulation of melanosomes in melanocytes, impaired movement of synaptic vesicles, potential inability of ER to move into the dendritic spines	dilute [39]
Griscelli Type 2	*RAB27A*	Hypopigmentation of skin and hair Immunological disorder * Increased bleeding (Ashen mice)	Abnormal accumulation of melanosomes in melanocytes, impaired natural-killer cell function, uncontrolled lymphocyte, and macrophage activation * Reduction in the number of platelet dense granules (Ashen mice).	ashen [40]
Griscelli Type 3	*MLPH*	Hypomelanosis with no immunological and neurological manifestation.	Perinuclear aggregation of melanosomes in melanocytes	leaden [41]
MYH9RD	*MYH9*	Macrothrombocytopenia with mild bleeding tendency, may develop kidney dysfunction, deafness, and cataracts	Enlarged platelets, Döhle-like bodies in the cytoplasm of neutrophils	R702C, D1424N, E1841K [42]
FHL3	*UNC13D*	Typical FHL with early onset of uncontrolled activation of T lymphocytes and macrophages, * bleeding	Lytic granules of NK cells and CTLs fail to fuse with the plasma membrane, * abnormalities in platelet secretion	Unc13d(Jinx) [43]
Chediak–Higashi Syndrome	*LYST*	Immunodeficiency, oculocutaneous albinism, bleeding, recurrent infections, neurologic defects, lymphohistiocytosis	Giant intracellular granules in different cell types including neurons, immune cells, melanocytes, platelets	beige [44]
HPS1 HPS4	*HPS1* *HPS4*	Restrictive lung disease, pulmonary fibrosis, granulomatous colitis, prolonged bleeding (variable), hypopigmentation (variable), inflammatory bowel disease, acanthosis nigricans and hypertrichosis.	Cells filled with phospholipids droplets; enlarged lamellar bodies; absence of platelet dense granules	pale ear [45] light ear [46]
HPS2	*AP3B1*	Immunodeficiency, neutropenia, recurrent infections, hypopigmentation, acanthosis nigricans and hypertrichosis, dysplastic ace tabulae, poor balance, conductive hearing loss, haemorrhage	Diminished amounts of *β*3A protein, melanocytes with abundant multivesicular structures, CTL with increased size of the endosomal network fail to induce polarization of lytic granules to the IS and decreased ability to kill targets, enlarged LB.	pearl [47]
HPS3	*HPS3*	Elevated creatinine clearance, ocular albinism, bruising and epistaxis	Mislocalisation of LAMP-1 and LAMP-3 as well as melanosome targeted proteins, decreased levels of melanin in melanocytes	cocoa [48]
HPS5	*HPS*5	Elevated creatinine clearance, nystagmus, bruising, hypercholesterolemia	Absent platelet dense bodies, LAMP-3 distributed in granules in a perinuclear region and not in the dendritic processes like normal.	ruby-eye2 [49]
HPS6	*HPS*6	T creatinine clearance, respiratory and urinary tract infections, epistaxis, bleeding, oculocutaneous albinism, urinary and rectal incontinence global developmental delay hearing loss	Mislocalisation of melanosome targeted proteins Deficient platelet dense granules	ruby-eye [49]
HPS7	*DTNBP1*	Oculocutaneous albinism, bleeding tendency, mild shortness of breath with exertion, reduced lung compliance	Abnormal melanosomes *	sandy [50]
HPS8	BLOC1S3	Incomplete oculocutaneous albinism, mild platelet dysfunction, easy bruising, hematomas after venesection, frequent epistaxis and prolonged bleeding after surgery or childbearing	Increased kidney lysosomal glycosidase activities, decreased platelet dense bodies, immature melanosomes, decreased melanin levels, abnormal intracellular tyrosinase distribution *	reduced pigmentation [51]
HPS9	*BLOC1S6*	Oculocutaneous albinism, nystagmus, recurrent cutaneous infections, thrombocytopenia, and leukopenia	NK cells have an increased surface levels of LAMP1 and CD63, decreased granulation, and decreased cytolytic activity	pallid [52]
HPS10	*AP3D1*	Immunodeficiency, oculocutaneous albinism, impaired hearing, severe neurologic impairment, delayed development and seizures, recurrent infections	Reduced levels of renal lysosomal enzymes, deficiency in the dense granules of platelets. Lack of AP3 in mocha tissues and reduced levels of the zinc transporter Znt3 in brain	mocha [53]
ARC	*VPS33B* *VIPAS39*	Arthrogryposis, renal tubule dysfunction neonatal cholestasis, ichthyosis, sensorineural deafness, central nervous system malformation, bleeding, recurrent infections, and severe failure to thrive.	Abnormal platelet α-granule and LB biogenesis, mislocalisation of apical proteins in hepatocytes, defects in collagen homeostasis.	Vps33b^fl/fl^ [54,55] Vipas39^fl/fl^ [56]

## Data Availability

The data presented in this study is available on request from the corresponding author.

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
