# Peer review of "Overlapping Machinery in Lysosome-Related Organelle Trafficking: A Lesson from Rare Multisystem Disorders"

_cells, 2022, doi:10.3390/cells11223702_

Round 1
Reviewer 1 Report
As a non-expert in LRO but basic cell biologist, I found this review very informative and well written. The illustrations and tables are very adequate to clarify the subject. My comments are limited to some typo’s that I encountered
Figure 1: Lysosme
Line 84: correct LOR
Line 201: should ‘In’ be removed?
Line 36As a non-expert in LRO but basic cell biologist, I found this review very informative and well written. The illustrations and tables are very adequate to clarify the subject. My comments are limited to some typo’s that I encountered
Figure 1: Lysosme
Line 84: correct LOR
Line 201: should ‘In’ be removed?
Line 369 should it be ‘vesicles’?
Line 382: should it be ‘affect’?
Line 457: ko : has this been defined?9 should it be ‘vesicles’?
Line 382: should it be ‘affect’?
Line 457: ko : has this been defined?
Author Response
As a non-expert in LRO but basic cell biologist, I found this review very informative and well written. The illustrations and tables are very adequate to clarify the subject. My comments are limited to some typo’s that I encountered
We thank the Reviewer for their positive feedback on the review.
Figure 1: Lysosme
Corrected
Line 84: correct LOR
Corrected (line 109)
Line 201: should ‘In’ be removed?
Corrected (line 358)
Line 369 should it be ‘vesicles’?
Corrected (line 639)
Line 382: should it be ‘affect’?
Corrected (lines 446)
Line 457: ko : has this been defined?9 should it be ‘vesicles’?
Corrected (lines 728-729)
Reviewer 2 Report
The manuscript titled “Overlapping Machinery in Lysosome-Related Organelle Trafficking: A Lesson from Rare Multisystem Disorders” by Banushi and Simpson provides a summary of hereditary rare multisystem disorders that arise from defective functions of lysosome-related organelles (LROs). The figures and table are very informative and give a snapshot review of both basic biological functions of LROs and disease mechanisms. The readability was, however, diminished significantly by poor organization and structure of the manuscript (see points below). Concerns regarding the last section which presented a hypothesis should be noted. While intriguing and conceivable, the hypothesis should be rigorously tested with proper analyses before being included in a peer-reviewed publication. Specific comments are as follows:
1. Line 84: Typo “LOR”.
2. Table 1: Gene nomenclature in the “Mutations” column should be standardized, e.g. MYO5A for Myosin V and UNC13D for Munc13-4. Human genes should be spelled all capitalized and italicized.
3. Line 134 & 137: “GS1 typical albinism” and “GS2 typical albinism” should be corrected to “Griscelli syndrome type 1 (GS1)” and “Griscelli syndrome type 2 (GS2)” respectively. Both types manifest hypopigmentation. Calling other distinct manifestations to be associated with albinism could be misleading.
4. Since hypopigmentation is the common phenotype of GS1/2/3, the authors can first summarize the role of RAB27A, MyoV, and MYH9 in melanosome trafficking, before describing the distinct phenotypes exhibited in individual GS types due to gene-specific function in neurons and immune cells.
5. The subheadings appear as a mix of statements, gene names, and disease. It is not unclear whether the following section is about basic biology, gene function, or a disease. They can be described more specifically and relevant to the diseases covered by this manuscript. For instance, “4. Myosins” can become “4. GS Type 1 caused by defective MYO5A-mediated LRO trafficking”; “6. Lyst is a common regulator of LROs fission” can become “6. CHS is caused by impaired LYST-mediated LRO fission”.
6. Line 254-259: Citation for the described immune cell phenotypes is missing. Editing is also needed to correct the sentence structure and fluency.
7. Section 8: Extensive reorganization is required. Although this section is all about AP3, it reads like an information dump without clear rule of paragraphing. One paragraph talks about the Golgi-to-vacuole function in yeast with no apparent relevance to disease phenotype. Another paragraph describes both platelet deficiency and neuronal phenotypes. What are the references for Line 325-326 and Line 331-333? The unconsidered presentation flow led to a frustrating read. Also, a discussion on why HPS2 and HPS10 are classified as two diseases and manifest non-overlapping phenotypes despite the common loss of AP3 function will be appreciated.
8. VPS33A/B and VIPAR are abbreviations for Vacuolar Protein Sorting-Associated Protein 33A/B and VPS33B-interacting Protein Involved in Polarity and Apical protein Restriction respectively. Also, the gene name for VIPAR, VIPAS39, should be used where genetic mutation or gene expression are concerned.
9. Line 420-432: What is the relevance of CORVET to ARC syndrome?
10. Line 457: Please specify what “Vps33b liver specific ko” means.
11. Section 11 and Figure 7: A multitude of caveats: 1) It is not clear whether the cited studies (Ref 9&150) demonstrated the “synergic expression at mRNA” levels of VPS33B and VIPAS39. 2) Using relative RNA levels to test for co-regulated expression of 2 genes is not appropriate. Such approach does not account for cell-to-cell variation even in a single cell type. At the very least, analyses of correlation within single cells, e.g. Pearson coefficient, are required to demonstrate 2 genes are co-expressed, or not. 3) Cell type-specific post-transcriptional processes were not considered. A relatively larger difference in mRNA levels could, for example, still yield comparable protein levels if the 2 proteins exhibit different rate of degradation. 4) The authors included VPS33A in the discussion but no meta-analysis was done on VPS33A.
Author Response
The manuscript titled “Overlapping Machinery in Lysosome-Related Organelle Trafficking: A Lesson from Rare Multisystem Disorders” by Banushi and Simpson provides a summary of hereditary rare multisystem disorders that arise from defective functions of lysosome-related organelles (LROs). The figures and table are very informative and give a snapshot review of both basic biological functions of LROs and disease mechanisms. The readability was, however, diminished significantly by poor organization and structure of the manuscript (see points below). Concerns regarding the last section which presented a hypothesis should be noted. While intriguing and conceivable, the hypothesis should be rigorously tested with proper analyses before being included in a peer-reviewed publication. Specific comments are as follows:
We thank the Reviewer for their positive comment on the review and the opportunity to organise better the structure of the manuscript.
We have restructured also the last section on the transcriptional analysis to avoid misinterpretation.
- Line 84: Typo “LOR”.
Corrected (line 109)
- Table 1: Gene nomenclature in the “Mutations” column should be standardized, e.g. MYO5A for Myosin V and UNC13D for Munc13-4. Human genes should be spelled all capitalized and italicized.
Corrected (Table 1)
- Line 134 & 137: “GS1 typical albinism” and “GS2 typical albinism” should be corrected to “Griscelli syndrome type 1 (GS1)” and “Griscelli syndrome type 2 (GS2)” respectively. Both types manifest hypopigmentation. Calling other distinct manifestations to be associated with albinism could be misleading.
Corrected (lines 263, 265)
- Since hypopigmentation is the common phenotype of GS1/2/3, the authors can first summarize the role of RAB27A, MyoV, and MYH9 in melanosome trafficking, before describing the distinct phenotypes exhibited in individual GS types due to gene-specific function in neurons and immune cells.
Corrected (lines 224-230)
- The subheadings appear as a mix of statements, gene names, and disease. It is not unclear whether the following section is about basic biology, gene function, or a disease. They can be described more specifically and relevant to the diseases covered by this manuscript. For instance, “4. Myosins” can become “4. GS Type 1 caused by defective MYO5A-mediated LRO trafficking”; “6. Lyst is a common regulator of LROs fission” can become “6. CHS is caused by impaired LYST-mediated LRO fission”.
We agree with the reviewer and we have changed the subheadings:
“4. Myosins” to “4. GS1 and MYH9RD are caused by defective Myosin-mediated LRO trafficking”
“5. Rab27a and its effectors” to “5. Rab27a and its effectors in LRO trafficking”
“6. Lyst is a common regulator of LROs fission” to “6. CHS is caused by impaired LYST-mediated LRO fission”.
“8. AP-3 sorts different cargoes in different LROs” to “8. AP-3 sorts different cargoes in different LROs and its deficient function associates with HPS2 and HPS10”
- Line 254-259: Citation for the described immune cell phenotypes is missing. Editing is also needed to correct the sentence structure and fluency.
Corrected (lines 433-437)
- Section 8: Extensive reorganization is required. Although this section is all about AP3, it reads like an information dump without clear rule of paragraphing. One paragraph talks about the Golgi-to-vacuole function in yeast with no apparent relevance to disease phenotype. Another paragraph describes both platelet deficiency and neuronal phenotypes. What are the references for Line 325-326 and Line 331-333? The unconsidered presentation flow led to a frustrating read. Also, a discussion on why HPS2 and HPS10 are classified as two diseases and manifest non-overlapping phenotypes despite the common loss of AP3 function will be appreciated.
We thank the Reviewer for the comment on section 8. We followed all the recommendations and completely changed the structure of this section and we believe that the presentation flow has now significantly improved.
Additionally, we added the references as per Reviewer’s recommendation, and a discussion on why HPS2 and HPS10 are classified as two diseases and manifest non-overlapping phenotypes despite the common loss of AP3 function (lines 501-507)
- VPS33A/B and VIPAR are abbreviations for Vacuolar Protein Sorting-Associated Protein 33A/B and VPS33B-interacting Protein Involved in Polarity and Apical protein Restriction respectively. Also, the gene name for VIPAR, VIPAS39, should be used where genetic mutation or gene expression are concerned.
Abbreviations corrected in the main text and Figure legend 7, Figure 7 (A-C) panels.
Line 420-432: What is the relevance of CORVET to ARC syndrome?
VPS33B and VIPAR are not part of the mammalian CORVET complex but they form a separate complex, therefore CORVET is not relevant to ARC syndrome (lines 693-703).
- Line 457: Please specify what “Vps33b liver specific ko” means.
Corrected (lines 728-729)
- Section 11 and Figure 7: A multitude of caveats: 1) It is not clear whether the cited studies (Ref 9&150) demonstrated the “synergic expression at mRNA” levels of VPS33B and VIPAS39. 2) Using relative RNA levels to test for co-regulated expression of 2 genes is not appropriate. Such approach does not account for cell-to-cell variation even in a single cell type. At the very least, analyses of correlation within single cells, e.g. Pearson coefficient, are required to demonstrate 2 genes are co-expressed, or not. 3) Cell type-specific post-transcriptional processes were not considered. A relatively larger difference in mRNA levels could, for example, still yield comparable protein levels if the 2 proteins exhibit different rate of degradation. 4) The authors included VPS33A in the discussion but no meta-analysis was done on VPS33A.
We added a paragraph (lines 624-634) that explains the co-regulation of VPS33B and VIPAS39 in ARC syndrome. We therefore think that these considerations justify the analysis reported in this section and address some of the Reviewer’s concerns.
The analysis reported in Figure 7 is not aimed at using relative RNA levels to test for co-regulated expression of 2 genes. We agree with the Reviewer that this is not appropriate and requires analyses of correlation within single cells (e.g. Pearson coefficient) and cell type-specific post-transcriptional processes should be considered.
In this figure we have simply plotted single cell RNA expression of VPS33B and VIPAS39 in different cell types that are already available from the Human Protein Atlas. We have modified the interpretations of this analysis to address the Reviewer’s concerns.
Our analysis could however offer interesting considerations for future experimental work that is needed to test the relationship between RNA expression and protein function within macromolecular complexes in different cell types, and therefore adds value to this review. This experimental work goes beyond the scope of this review.
An additional relevance to this analysis is related to its intrinsic results showing how the majority (~65%) of the cell types exhibit a similar expression in (1<fold change<2). Given the high number of cell types analysed, the probability that these results are due to chance are very low. From literature, CHEVI complex has already been shown to exist as a separate binary complex and its members VPS33B and VIPAR show synergic expression. Taken together, these considerations suggest that our analysis is relevant to this specific example here reported.
Reviewer 3 Report
This is a review article about the trafficking machinery of the lysosome-related organelles (LROs). The authors have summarized the mechanisms of trafficking into various LROs. In addition, they have discussed the aspects of rare multisystem disorders that are caused by mutations in genes coding for the proteins of this trafficking machinery. This is a well-written and important contribution to the field. I have a couple of remarks that should help to finalize the paper for publication.
1) My main remarks concerns the transcriptional analysis data shown in Figures 6 and 7. The editor should check if primary data are allowed in a review article. If not, this section can be deleted without loss of importance of the manuscript.
2) Legend of Figure 1 is too exhaustive. Some of the text (explanations of trafficking steps) can be transferred to the main text.
3) Lines 276-278: please rewrite the sentence. (A summary… are summarized)
4) Figure 5 is too small. The text in 5A is barely eligible in this size.
5) HPS is listed twice in the abbreviations.
Author Response
This is a review article about the trafficking machinery of the lysosome-related organelles (LROs). The authors have summarized the mechanisms of trafficking into various LROs. In addition, they have discussed the aspects of rare multisystem disorders that are caused by mutations in genes coding for the proteins of this trafficking machinery. This is a well-written and important contribution to the field. I have a couple of remarks that should help to finalize the paper for publication.
We thank the Reviewer for their positive feedback on the review.
1) My main remarks concerns the transcriptional analysis data shown in Figures 6 and 7. The editor should check if primary data are allowed in a review article. If not, this section can be deleted without loss of importance of the manuscript.
We ask the editor for their input on this concern. However, the data shown in Figure 7 might not be classified as primary data, given its free availability for the Human Protein Atlas. Experimental work is required to confirm the hypothesis here reported.
2) Legend of Figure 1 is too exhaustive. Some of the text (explanations of trafficking steps) can be transferred to the main text.
Adjusted accordingly.
3) Lines 276-278: please rewrite the sentence. (A summary… are summarized)
Corrected to ‘A summary of the clinical and cellular phenotype for each HPS subtype is reported in Table 1’ (line 455-456)
4) Figure 5 is too small. The text in 5A is barely eligible in this size.
Corrected.
5) HPS is listed twice in the abbreviations.
Corrected.
Round 2
Reviewer 2 Report
Numerous typos still exist throughout the manuscript. Please review and edit carefully.